# Developing a recombinase-aided amplification method combined with a lateral flow dipstick assay for rapid triplex detection of bovine coronavirus, infectious bovine rhinotracheitis virus, and bovine viral diarrhea virus

Yingcai Ma,[1,2] Xinhao Wang,[1,2] Rulong Chen,[3] Keer Dong,[1,2] Xiaojie Yu,[1,2] Na Li,[1,2] Li Yang,[1,2] Qi Zhong,[4] Gang Yao,[1,2] Xuelian Ma[1,2]

**ABSTRACT** To establish a sensitive, simple rapid method for detecting bovine coronavirus (BCoV), infectious bovine rhinotracheitis virus (IBRV), and bovine viral diarrhea virus (BVDV). Recombinase-aided amplification (RAA) was used to amplify template DNA or cDNA, and a lateral flow dipstick (LFD) was used to interpret the results after amplification was completed. Seventy-three rectal and nasal bovine samples were tested to evaluate the performance of the RAA-LFD assay and were tested in parallel via polymerase chain reaction (PCR) and reverse transcription quantitative PCR (RT-qPCR) for comparison. The triplex RAA-LFD assay was completed within 20 min at 39°C. This method demonstrated reasonable specificity, with no cross-reactivity to bovine norovirus (BNoV), bovine rotavirus (BRV), bovine parainfluenza virus 3 (BPIV3), or bovine respiratory syncytial virus (BRSV). The detection thresholds for IBRV, BVDV, and BCoV are $3.20 \times 10^3$ copies/µL, $2.21 \times 10^3$ copies/µL, and $4.13 \times 10^3$ copies/µL, respectively. Triplex RAA-LFD practicality was tested on 73 rectal and nasal bovine swabs from diarrheic cattle on commercial farms. Compared with RT-qPCR, the triplex RAA-LFD assay yielded a clinical sensitivity of 98.41%, 80.00%, and 85.71%, a positive predictive value (PPV) of 100.00%, and kappa coefficients of 0.94, 0.88, and 0.91 for BCoV, IBRV, and BVDV, respectively. Compared with PCR, it achieved 100.00%, 75.00%, and 87.50% sensitivity; 41.94%, 75.00%, and 58.33% PPV; and 0.33, 0.73, and 0.65 kappa coefficients for BCoV, IBRV, and BVDV, respectively. A triplex RAA-LFD method was developed to simultaneously detect BCoV, IBRV, and BVDV, offering an efficient solution for identifying multiple cattle viral pathogens.

**IMPORTANCE** This study developed a triplex RAA-LFD assay for the simultaneous detection of BCoV, IBRV, and BVDV. The method demonstrated high sensitivity (detection limits: $10^3$ copies/µL), specificity (no cross-reactivity with related viruses), and speed (20 min, including amplification and visualization). Clinical validation with 73 bovine samples showed strong agreement with RT-qPCR (kappa coefficients: 0.88–0.94), supporting its reliability. Compared with that of PCR, the sensitivity of this method remains high although the PPV varies. This rapid, low-cost, field-deployable assay improves surveillance of co-infections in cattle, aiding timely diagnosis and disease management, particularly in resource-limited settings.

**KEYWORDS** bovine coronavirus, infectious bovine rhinotracheitis virus, bovine viral diarrhea virus, triplex recombinase-aided amplification, lateral flow dipstick

B eef cattle production is one of the most important pillar industries in Xinjiang's animal husbandry. With the expansion of the scale of cattle breeding in Xinjiang,

Address correspondence to Xuelian Ma, maxuelian@xjau.edu.cn.

The authors declare no conflict of interest.

See the funding table on p. 14.

the prevention and control of beef cattle diseases have become increasingly challenging for safe and healthy production in the beef cattle industry. Viral infection is one of the most important causes of morbidity and mortality in cattle (1–3). Three of the most common viruses affecting cattle health are bovine coronavirus (BCoV), infectious bovine rhinotracheitis virus (IBRV), and bovine viral diarrhea virus (BVDV) (4). BCoV is a single-stranded positive-sense RNA virus that can cause respiratory disease and diarrhea in calves and winter dysentery in adult cattle (5, 6). The latest studies have indicated that the prevalence of BCoV ranges from 3.4% to 69.00% in diarrheal feces and 11.8% to 74.60% in BRD-associated symptoms (7). IBRV is a double-stranded DNA virus known as bovine alphaherpesvirus 1 (BoHV-1), which can cause upper respiratory tract infections, conjunctivitis, and genital disorders and decrease fertility and abortions in cattle; IBRV seroprevalence ranges from 15.60% to 55.49% (8–10). BVDV is an enveloped single-stranded positive-sense RNA virus. BVDV infection in cattle results in symptoms such as diarrhea, respiratory issues, and reproductive and immunological dysfunctions. It can also infect the fetus through vertical transmission, potentially causing abortion or stillbirth during pregnancy. The seroprevalence of BVDV infection in cattle is high, with rates approaching 52% (11–13). These viral infections in cattle cause significant financial losses for the beef industry. At present, the early detection of BCoV, IBRV, and BVDV is challenging, leading to a lag in disease prevention and control. Therefore, early detection and early treatment are the keys to prevention and control.

The traditional detection methods for BCoV, IBRV, and BVDV include virus isolation and identification (14, 15), PCR (16), reverse transcriptase quantitative PCR (RT-qPCR) (17), and enzyme-linked immunosorbent assays (ELISAs) (18). Despite the good specificity and sensitivity of these methods, the dependency on the operating environment, equipment, and personnel quality limits their ability to test samples in the field, which is laborious and time-consuming. Simple, convenient, accurate, rapid, and visual detection technology is essential for the prevention and control of viral diseases.

Recombinase-aided amplification (RAA) is a technique for nucleic acid amplification involving the use of recombinant enzymes, single-stranded DNA-binding proteins, and DNA polymerase under isothermal conditions (19). Recombinant enzymes, single-stranded DNA-binding proteins, and primer formation complexes scan double-stranded DNA, unwinding double-stranded DNA at the sequence homologous to the primer, and single-stranded DNA-binding proteins prevent single-stranded DNA renaturation and complete chain exponential extension by DNA polymerase in the presence of dNTPs in 5–15 min. Then, via antigen- and antibody-specific recognition, the amplification products are visualized via a lateral flow dipstick (LFD), and the results are directly observed with the naked eye within 5 min (20). The combination of RAA-LFD is an innovative, simple, and user-friendly method for analyzing various common pathogens in the field; this method has been widely used in the field to detect parasites (21), viruses (22–24), and other pathogens (20). Currently, RAA-LFD is mainly used in the detection of a single or two viruses. However, the RAA-LFD method for simultaneously detecting BCoV, IBRV, and BVDV has not been reported.

This study was conducted to develop a triplex RAA-LFD assay with a certain sensitivity, repeatability, and specificity for the simultaneous detection of BCoV, IBRV, and BVDV. These findings can provide necessary data for more predictive and preventative actions to make important contributions to the rapid field diagnosis of common pathogens.

## MATERIALS AND METHODS

### Sample collection

From 2024 to 2025, 73 rectal and nasal swabs (50 rectal swabs and 23 nasal swabs) were obtained from diarrheic cows on commercial bovine farms in Kashgar, Yili, and Changji in Xinjiang Province, China. The samples were stored in liquid nitrogen until testing. All

experiments were approved by the Animal Welfare and Ethics Committee of Xinjiang Agricultural University (ethical clearance number 2024047).

The nucleic acids extracted from bovine norovirus (BNoV) (25), bovine rotavirus (BRV) (26), bovine parainfluenza virus 3 (BPIV3) (27), and bovine respiratory syncytial virus (BRSV) (28) were used as competitive positive controls in the assay.

## Nucleic acid extraction

For viral RNA extraction of BCoV and BVDV from the rectal swabs, the samples were vortexed in 2 mL of sterile phosphate-buffered saline (PBS, pH 7.4) and centrifuged at 10,000 rpm for 10 min at 4°C. Five hundred microliters of the supernatant was collected for viral RNA extraction via the use of 1 mL of TRIzol Reagent (Invitrogen, Waltham, USA), and the viral RNA was reverse transcribed to cDNA via a TransScriptIV Reverse Transcriptase Kit (TransGen, Beijing, China) following the manufacturer's instructions. IBRV viral DNA was extracted from nasal swabs with a TIANamp Genomic DNA Kit (Tiangen, Beijing, China) according to the manufacturer's instructions. The DNA/cDNA was subsequently resuspended in 50 µL of ddH$_2$O and stored at −80°C until use.

## Construction of positive control plasmids

In accordance with the sequences of the N gene, gB gene, and 5′-UTR gene of the BCoV, IBRV, and BVDV viruses published in GenBank, primers for PCR were designed via the primer design tool Primer 5 and synthesized by Shanghai Sangon Biotechnology Co., Ltd. (Table S2), which were used for amplification of the N gene, gB gene, and 5′-UTR gene of the BCoV, IBRV, and BVDV viruses from the extracted DNA/cDNA previously extracted. After amplification, the PCR product was purified via a TIANgel Purification Kit (Tiangen, Beijing, China) following the manufacturer's protocol. The purified product was ligated with the pEASY-T1 cloning vector (TransGen, Beijing, China) and subsequently transformed into DH5α competent cells. After the recombinant plasmid was amplified in Luria-Bertani (LB) medium, the plasmid was extracted via a small amount of Plasmid Extraction Kit (Tiangen, Beijing, China). Finally, the BCoV, IBRV, and BVDV recombinant plasmids were washed with 50 µL of ddH$_2$O, the plasmid concentration was quantified via a NanoDrop (NanoDrop Technologies, USA) and converted to copy number with the following formula: DNA copy number (copies/µL) = (6.02 × 10$^{23}$) × (DNA concentration (ng/µL) × 10$^{−9}$)/(DNA length × 660). The DNA was stored at −20°C.

Positive plasmids for bovine coronavirus (BCoV) (GenBank: MW711287.1), infectious bovine rhinotracheitis virus (IBRV) (GenBank: MK654723.1), and bovine diarrhea virus (BVDV) (GenBank: KF501393.1) were obtained from our laboratory.

## Design and screening of RAA primers and probes

On the basis of the principles of RAA primer and probe design (29), primary, secondary, and tertiary RAA primers and probes were designed to target different gene sequences of BCoV, IBRV, and BVDV via Primer Premier 5.0 software. These primers included 14 pairs of BCoV primers, 10 pairs of IBRV primers, and 14 pairs of BVDV primers (Table S1), which were synthesized by Sangon Biotech Co., Ltd. (Shanghai, China). Briefly, the secondary primers were designed on the basis of the optimum primary primers by shifting 1, 2, or 3 bp to both sides and fixing upstream primers to screen downstream primers. The tertiary primers were designed on the basis of the optimum secondary primers by increasing or reducing 1 or 2 bp on both sides. For each probe, the 5′ end was labeled with a FAM fluorophore, and a tetrahydrofuran (THF) site was placed in the middle of the probe, while the 3′ end was blocked by a C3-spacer phosphorylation spacer. Moreover, the 5′ ends of the downstream primers of the optimum tertiary primers were labeled with biotin, Tamra, or digoxin. The optimum tertiary primers and probes for RAA-LFD designed in this study are listed in Table S3. For the triplex RAA-LFD assay, the specific RAA-targeted gene associated with different reporter molecules can be captured by different detection antibodies, such as anti-biotin monoclonal antibodies, anti-tamra

monoclonal antibodies, and anti-digoxin monoclonal antibodies, which are deposited upon the T1, T2, and T3 test lines of multiplex LFDs via immunoassays.

## Preparations of LFD

The LFD test consists of a sample pad, a conjugate pad, a nitrocellulose（NC） membrane, and an absorbent pad. The colloidal gold solution was labeled with a mouse anti-FAM monoclonal antibody and sprayed evenly on the conjugate pad. The C line was coated with goat anti-mouse IgG; the T1 line was coated with an anti-biotin monoclonal antibody for the detection of BCoV; the T2 line was coated with an anti-tamra monoclonal antibody for the detection of IBRV; and the T3 line was coated with an anti-digoxin monoclonal antibody for the detection of BVDV. The test paper was assembled and cut into 2.5 mm strips for storage in a dry environment until use. These LFDs were provided by Shenzhen Zhenrui Biotechnology Co., Ltd.

## RAA-LFD assay

The RAA-LFD assay was performed using a commercial RNA isothermal rapid amplification kit (Amplification Future, Jiangsu, China) in a reaction volume of 50 µL. The reaction mixtures contained 20 µL C buffer, 5 µL L buffer, 2 µL of the forward primer (10 µM), 2 µL of the labeled reverse primer (10 µM), 0.6 µL of the probe (10 µM), 1.5 µL dNTPs (10 mM), 12 µL PR-core, 0.6 µL N-core, 3.8 µL of extracted template, and 2.5 µL B buffer. After all the ingredients were added to the reaction mixture, the mixture was shaken repeatedly upside down and mixed well 10 times. The reaction mixture was typically incubated for 15 min at 39℃. The amplification product was then diluted 10 times with phosphate-buffered saline and tested via LFD (Shenzhen Zhenrui Biotechnology Co., Ltd., Shenzhen, China). Both the test line and control line appear simultaneously indicating a positive result, whereas only the control line appears to indicate a negative result.

## Optimization of primer and probe concentrations

In accordance with the manufacturer's instructions for the use of the RAA kit, we designed four sets of reaction systems with different ratios of primer and probe. In the first set, the primer and probe of BCoV, IBRV, and BVDV were 0.150 µM (probe 0.040 µM), 0.200 µM (probe 0.060 µM), and 0.100 µM (probe 0.020 µM), respectively. In the second set, the primer and probe of BCoV, IBRV, and BVDV were 0.200 µM (probe 0.040 µM), 0.150 µM (probe 0.060 µM), and 0.050 µM (probe 0.020 µM), respectively. In the third set, the primer and probe and probes of BCoV, IBRV, IBRV, and BVDV were 0.180 µM (probe 0.040 µM), 0.180 µM (probe 0.050 µM), and 0.050 µM (probe 0.020 µM), respectively. In the fourth set, the primer and probe of BCoV, IBRV, and BVDV were 0.135 µM (probe 0.040 µM), 0.135 µM (probe 0.040 µM), and 0.135 µM, respectively.

## Optimization of the reaction time and temperature

The RAA-LFD reaction could be performed at various reaction times and temperatures. Thus, the optimal duration and temperature were examined on the basis of the optimal primer-probe concentration ratio. Six reaction times (min), 11, 12, 13, 14, 15, 16, 17, and 18, and six temperatures (℃), 37, 38, 39, 40, 41, and 42, were designed for the optimal condition assessment. All optimization experiments were performed with 1 µL of BCoV ($4.13 \times 10^7$ copies/µL), 1 µL of IBRV ($3.20 \times 10^7$ copies/µL), and 1 µL of BVDV ($2.21 \times 10^7$ copies/µL) mixture in the reaction as templates.

## Specificity and sensitivity of the triplex RAA-LFD system

The *in vitro*-transcribed cDNA of BCoV and BVDV or IBRV DNA-positive plasmids were quantified and adjusted to $10^7$ copies/µL. The positive controls then underwent specific reactions in single, double, and triplex mixtures. Moreover, the positive plasmids of BCoV

$(4.13 \times 10^7$ copies/µL), 1 µL of IBRV $(3.20 \times 10^7$ copies/µL), and 1 µL of BVDV $(2.21 \times 10^7$ copies/µL) were diluted to $10^5$, $10^3$, $10^2$, and $10^1$ copies/µL, respectively, to determine the detection limit of the triplex RAA-LFD assay.

## Testing of clinical samples via the triplex RAA-LFD assay

Following the manufacturer's instructions, for 73 rectal and nasal swabs (50 rectal swabs and 23 nasal swabs), the viral RNA of BCoV and BVDV and the DNA of IBRV were extracted via the FastPure Viral DNA/RNA Mini Kit (Vazyme, Nanjing, China). The DNA/RNA mixture was stored at −20°C until testing. The results of the triplex RAA-LFD assay were compared with those of the RT-qPCR and PCR methods. The degree of agreement between the triplex RAA-LFD and RT-qPCR or PCR assay results was measured according to the specificity, sensitivity, kappa value, positive predictive value (PPV), and negative predictive value (NPV) of each assay. The calculation method was obtained from the literature (30, 31). The coincidence rate (CR) between the two methods was calculated with the following formula: ([number of positive samples detected with both methods + number of negative samples detected with both methods]/total number of samples) × 100%.

## RESULTS

### Triplex RAA-LFD assay strategy

The N gene, gB gene, and 5′UTR gene are the targets for the diagnosis of three viruses: BCoV, IBRV, and BVDV, respectively. The chemical modification of the RAA reaction components enables different labeling of the target gene fragment, thus facilitating differential diagnosis between three targets on the lateral flow analysis strip with multiple detection lines (Fig. 1). Specifically, to diagnose BCoV, the fragment was amplified with a forward primer, a FAM-labeled probe, and a biotin-labeled reverse primer in an RAA reaction. The specific target gene fragment has Fam and Biotin labels at both 5′ ends. Similarly, the fragment was amplified, and the specific target gene fragment had FAM-Tamra and FAM-Digoxin labels at both 5′ ends for diagnosing IBRV and BVDV infections (Fig. 1). Three sets of primers/probes for the diagnosis of BCoV, IBRV, and BVDV were added to the same RAA reaction mixture and optimized so that the cDNA/DNA template of each virus could be explicitly amplified with the corresponding markers required (Fig. 1). For amplified result readings, the LFD strip with three test lines (T1/T2/T3 line) was used in addition to the control line (C line) (Fig. 1). On the LFD strip, the FAM-labeled target gene fragment binds to the gold particle-labeled mouse anti-FAM mAb when the target gene fragment moves on the conjugated pad. With further migration, Tamra-, Biotin-, and Digoxin-labeled target gene fragments generated via the use of the recombinant gene for each target virus as template are captured by the corresponding mAbs on the T lines. The uncaptured gold particle-labeled mouse anti-FAM mAb continues to move and is captured by the goat anti-mouse IgG on the C line (Fig. 1). Therefore, this triplex RAA-LFD assay could detect three viruses simultaneously.

### Screening of the optimal primers for the RAA-LFD assay

The AGE results of the primary RAA primers revealed that the BCoV-Forward primer 3/Reverse primer 3 (F3/R3) primer for the N gene, the IBRV-F1/R1 primer for the gB gene, and the BVDV-F3/R3 primer for the 5′UTR gene combination generated a specific band with high intensity and good amplification efficiency (Fig. S1). The AGE results of the secondary RAA primers revealed that the BCoV-F3/R6 primer for the N gene, the IBRV-F1/R1 primer for the gB gene, and the BVDV-F3/R3 primer for the 5′UTR gene combination generated a specific band with high intensity and good amplification efficiency (Fig. S2). The AGE results of the tertiary RAA primers revealed that the BCoV-F03/R03 primer for the N gene, the IBRV-F1/R1 primer for the gB gene, and the BVDV-F04/R04 primer for the 5′UTR gene combination generated a specific band

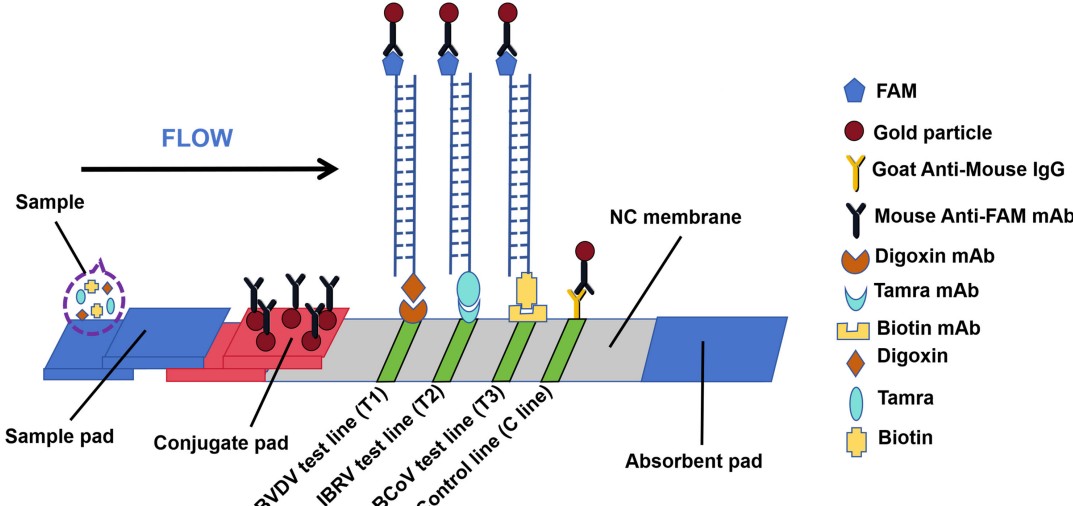

**FIG 1** Schematic of simultaneous lateral flow detection of triple targets. The FAM-labeled target gene fragment binds to the gold particulate-labeled mouse anti-FAM mAb, and the other end of the dual-labeled target gene is captured at one of the three detection zones.

with high intensity and good amplification efficiency. The other primer sets successfully amplified target sequences but with lower specificities or fewer amplification products (Fig. 2). Therefore, the primer pairs BCoV-F03/R03, IBRV-F1/R1, and BVDV-F04/R04 were selected as the optimum for subsequent experiments.

## Specificity of the singleplex RAA-LFD assay

To verify the feasibility of the singleplex RAA-LFD assay, the downstream primers of the N gene, gB gene, and 5′ UTR gene of BCoV, IBRV, and BVDV, respectively, were labeled with biotin, and the probes of BCoV, IBRV and BVDV, respectively, were labeled with FAM and reacted with positive plasmids. The results revealed that BCoV-, IBRV-, and BVDV-positive control plasmids presented specific bands on the test line (T line) of the corresponding test strip (Fig. 3A), which indicated that the positive plasmid and RAA-LFD assay of BCoV, IBRV, and BVDV could be used for subsequent tests.

The results showed that the RAA-LFD assay specifically detected BCoV, IBRV, and BVDV, and there was no cross-reactivity with other viruses (Fig. 3B through D). The above results indicated that the RAA-LFD assay of BCoV, IBRV, and BVDV has good uniplex detection specificity.

## The screening of downstream markers and establishment of the triplex RAA-LFD assay

To establish a good triplex RAA-LFD system, on the basis of the biotin-labeled down-stream primers of BCoV, IBRV, and BVDV, we further added two downstream primers labeled with Tamra and Digoxin. Moreover, we used LFDs labeled with Biotin, Tamra, and Digoxin for downstream marker screening and cross-specificity detection, respectively. The results revealed that biotin-labeled BCoV, tamra-labeled IBRV, and digoxin-labeled BVDV presented the best binding ability (Fig. 4A through C). Moreover, only biotin-labeled BCoV, tamra-labeled IBRV, and digoxin-labeled BVDV were specifically bound to their respective monoclonal antibody test strips (Fig. 4D through F). Furthermore, the triplex RAA-LFD assay revealed that T1, T2, and T3 test lines coated with biotin, tamra, and digoxin monoclonal antibodies, respectively, can specifically recognize the downstream markers biotin, Tamra, and digoxin of BCoV, IBRV, and BVDV, thereby generating specific bands (Fig. 4G).

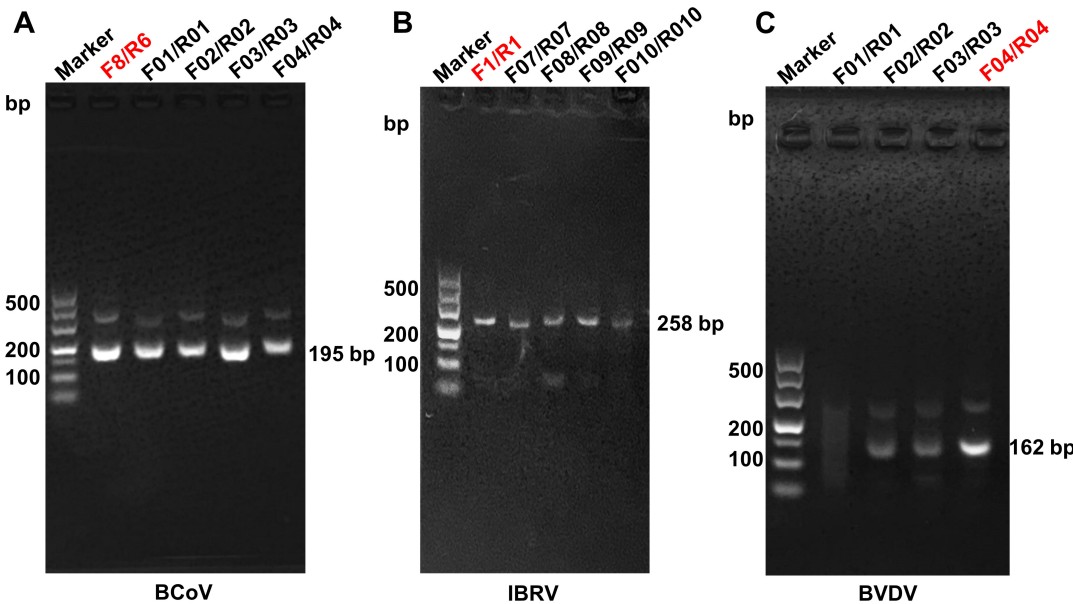

**FIG 2** Screening of optimal primers for the RAA assay. (A) RAA amplification of BCoV. Lane M is a 50–500 bp ladder marker, while lanes 1–5 are the results of RAA amplification of BCoV via the primers RAA-F8/R6, RAA-F01/R01, RAA-F02/R02, RAA-F03/R03, and RAA-F04/R04. (B) RAA amplification of IBRV. Lane M is a 50–500 bp ladder marker, while lanes 1–5 are the results of RAA amplification of IBRV via the primers RAA-F1/R1, RAA-F07/R07, RAA-F08/R08, RAA-F09/R09, and RAA-F010/R010. (C) RAA amplification of BVDV. Lane M is a 50–500 bp ladder marker, while lanes 1–4 are the results of RAA amplification of BVDV via the primers RAA-F01/R01, RAA-F02/R02, RAA-F03/R03, and RAA-F04/R04.

## Optimization of the reaction conditions for the triplex RAA-LFD assay

To explore the optimal reaction concentration, reaction time, and reaction temperature of the triplex RAA-LFD system, we optimized the reaction concentration, reaction time, and reaction temperature for the triplex RAA-LFD system. The reaction concentration results revealed that when the concentrations of the BCoV, IBRV, and BVDV primers were 0.135 µM and the probe concentration was 0.040 µM, compared with those of the other concentration groups, bright specific bands appeared in lines C and T of BCoV, IBRV, and BVDV (Fig. 5A). The results of the reaction times revealed that under the conditions described above, compared with those at the other time points, bright specific bands appeared in lines C and T of BCoV, IBRV, and BVDV at 15 min (Fig. 5B). Compared with those at other temperatures, the C line and the T line of BCoV, IBRV, and BVDV presented bright specific bands at 39°C (Fig. 5C).

## Results of the specificity and sensitivity of the triplex RAA-LFD assay

To further verify the assay specificity of the triplex RAA-LFD assay, we used the optimized reaction system. According to the specific reaction results, there was no cross-reaction between BCoV, IBRV, and BVDV (Fig. 6A).

To further verify the assay sensitivity of the triplex RAA-LFD assay, the positive BCoV, IBRV, and BVDV plasmids were diluted from $4.13 \times 10^7$ copies/µL, $3.20 \times 10^7$ copies/µL, and $2.21 \times 10^7$ copies/µL to $4.13 \times 10^1$ copies/µL, $3.20 \times 10^1$ copies/µL, and $2.21 \times 10^1$ copies/µL, respectively, via the optimized reaction system. The results of the sensitivity test revealed that the detection thresholds for BCoV, IBRV, and BVDV were $4.13 \times 10^3$ copies/µL, $3.20 \times 10^3$ copies/µL, and $2.21 \times 10^3$ copies/µL, respectively (Fig. 6B).

## Evaluation of clinical samples

To verify the clinical applicability of the triplex RPA-LFD assay, the practicality and efficiency of the triplex RAA-LFD were compared to those of RT-qPCR and PCR, and the sample test statistics are presented in Table S4. The results revealed that triplex

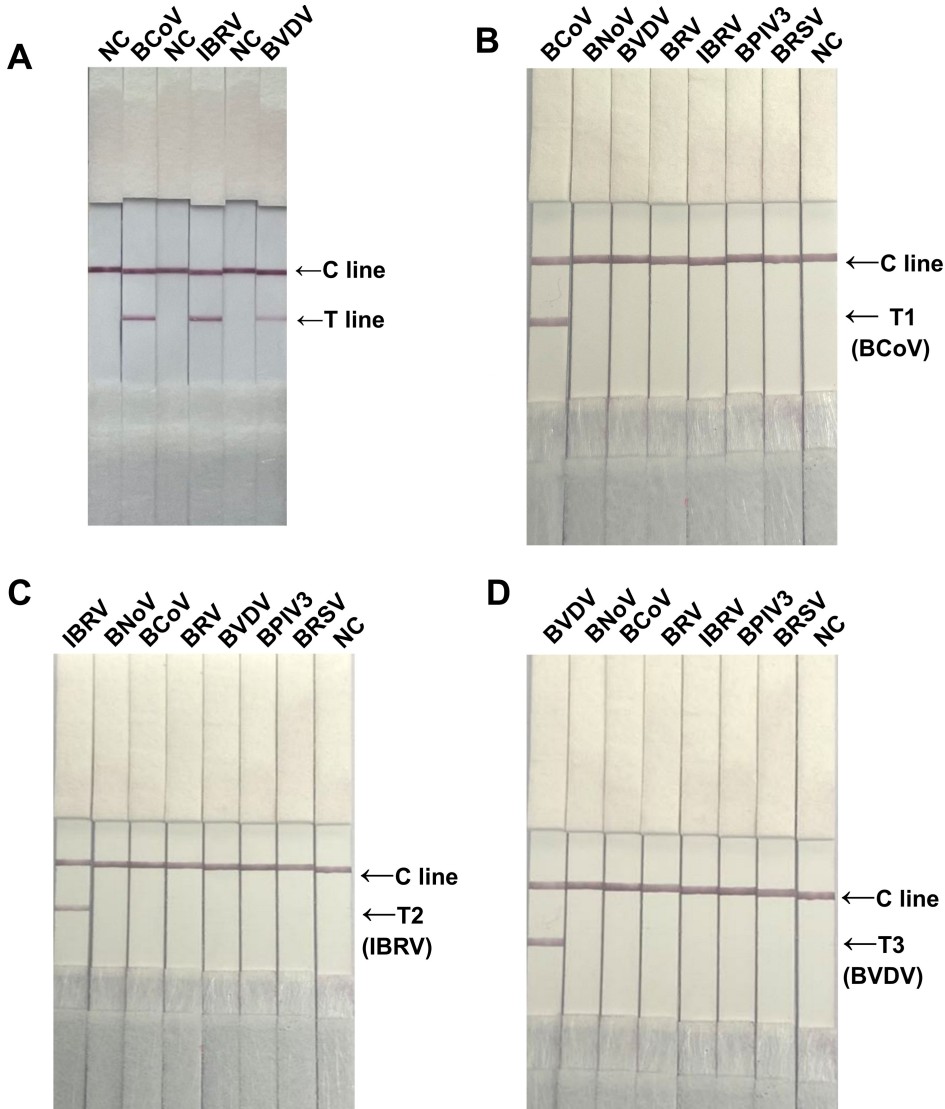

**FIG 3** Specificity detection of the singleplex RAA-LFD assay. (A) The feasibility of the singleplex RAA-LFD assay for BCoV, IBRV, and BVDV. Lane NC: negative control; Lanes 2, 4, and 6: positive plasmids for BCoV, IBRV, and BVDV, respectively. (B) Specificity of the singleplex RAA-LFD assay for BCoV. Lanes 1 to 7: BCoV, BNoV, BVDV, BRV, IBRV, BPIV3, and BRSV; Lane NC: Negative control. (C) Specificity of the singleplex RAA-LFD assay of IBRV. Lanes 1 to 7: IBRV, BNoV, BCoV, BRV, BVDV, BPIV3, and BRSV; Lane NC: Negative control. (D) Specificity of the singleplex RAA-LFD assay for BVDV. Lanes 1 to 7: BVDV, BNoV, BCoV, BRV, IBRV, BPIV3, and BRSV; Lane NC: Negative control.

RAA-LFD had a positive rate similar to that of the RT-qPCR assay [positive rates of 84.93% (62/73) and 86.30% (63/73), 5.48% (4/73) and 6.85% (5/73), and 16.44%, (12/73) and 19.18% (14/73) for BCoV, IBRV, and BVDV, respectively (Table 1)]. The coincidence rates of BCoV, IBRV, and BVDV were 98.63% (72/73), 98.63% (72/73), and 97.26% (71/73), the kappa values were 0.94, 0.88, and 0.91, and 98.41%, 80.00%, and 85.71% sensitivity, respectively, was achieved. Compared with those of PCR, the coincidence rates of BCoV, IBRV, and BVDV were 50.68% (37/73), 97.26% (71/73), and 91.78% (67/73), the kappa values were 0.33, 0.73, and 0.65, and 100.00%, 75.00%, and 87.50% sensitivity were achieved, respectively (Table 2). Overall, these data suggest that the triplex RAA-LFD assay has better sensitivity and specificity than PCR does. In terms of the time efficiency of the three detection methods, triplex RAA-LFD takes less time than RT-qPCR and PCR, which can shorten the detection time by half.

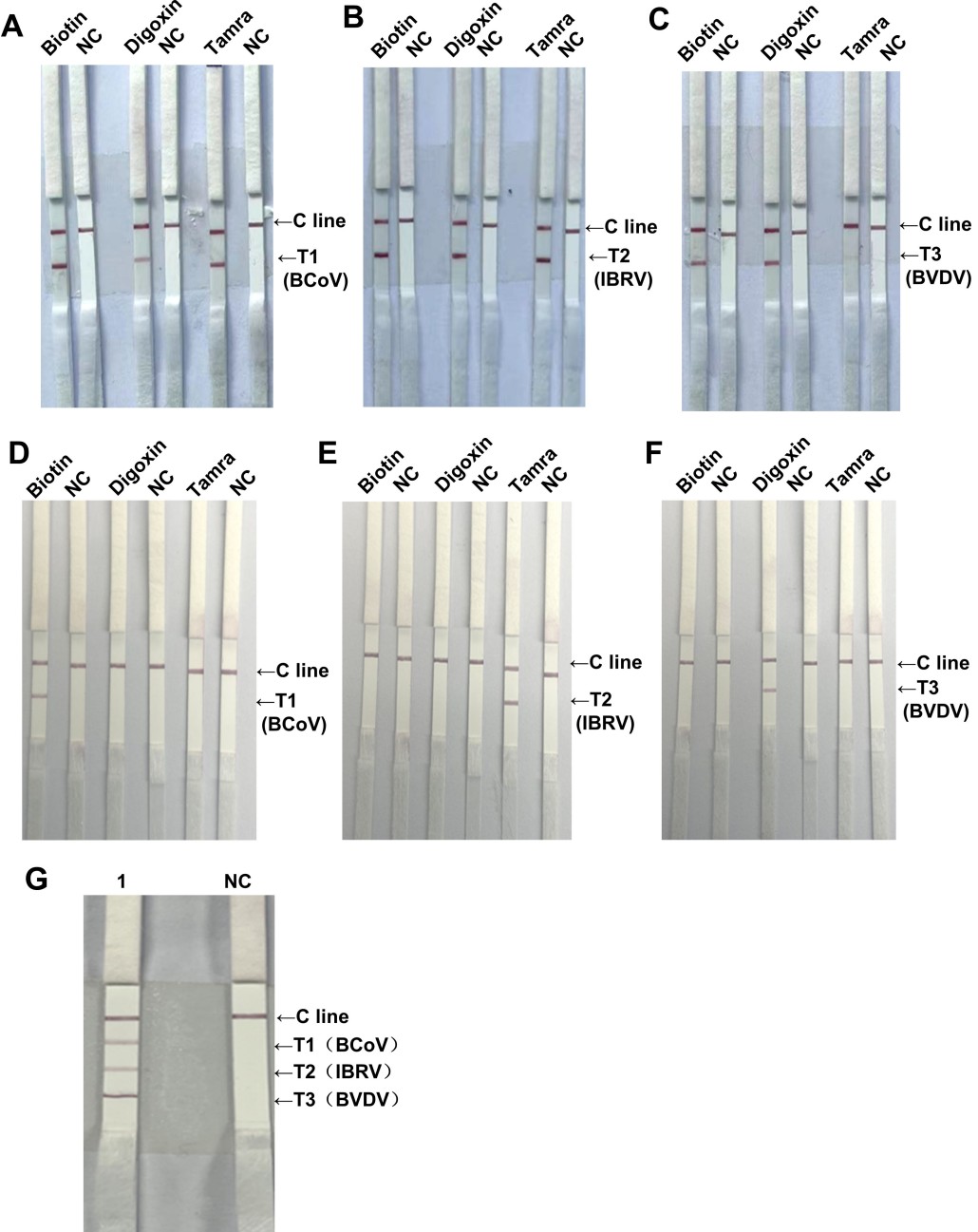

**FIG 4** The screening of downstream markers and establishment of the triplex RAA-LFD system. (A–C) Screening of downstream markers of different pathogens. (A) Lanes 1, 3, and 5: LFDs labeled with biotin, digoxin, and Tamra, respectively; the downstream primers of BCoV were labeled with biotin, digoxin, and Tamra, respectively; Lane NC: Negative control. (B) Lanes 1, 3, and 5: LFDs labeled with biotin, digoxin, and Tamra, respectively; the downstream primers of IBRV were labeled with biotin, digoxin, and Tamra, respectively; Lane NC: Negative control. (C) Lanes 1, 3, and 5: LFDs labeled with biotin, digoxin, and Tamra, respectively; the downstream primers for BVDV were labeled with biotin, digoxin, and Tamra, respectively; Lane NC: Negative control. (D–F) Cross-specificity detection of different pathogens. (D) Lanes 1 and 2: LFD labeled with biotin; Lanes 3 and 4: LFD labeled with digoxin; Lanes 5 and 6: LFD labeled with Tamra; Lanes 1–6: downstream primers of BCoV labeled with biotin; Lane NC: Negative control. (E) Lanes 1 and 2: LFD labeled with biotin; Lanes 3 and 4: LFD labeled with digoxin; Lanes 5 and 6: LFD labeled with Tamra; Lanes 1–6: downstream primers of IBRV labeled with Tamra; Lane NC: Negative control. (F) Lanes 1 and 2: LFD labeled with biotin; Lanes 3 and 4: LFD labeled with digoxin; Lanes 5 and 6: LFD labeled with Tamra; Lanes 1–6: downstream primers of BVDV labeled with digoxin; Lane NC: Negative control. (G) Triplex RAA-LFD assay. Lane 1: LFDs coated with biotin, Tamra, and digoxin monoclonal antibodies. The T1, T2, and T3 test lines can specifically recognize the downstream markers Biotin, Tamra, and Digoxin of BCoV, IBRV, and BVDV, respectively; Lane NC: Negative control.

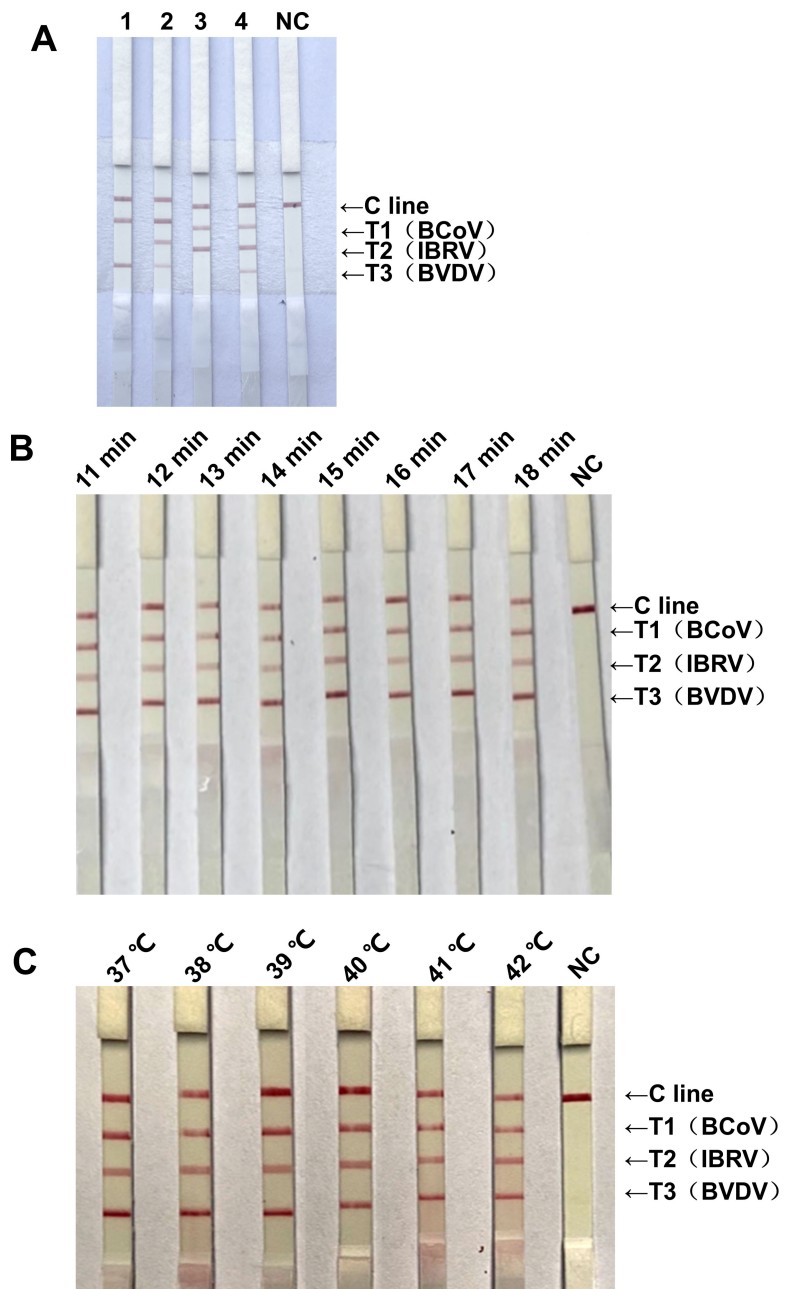

FIG 5 Optimization of the reaction conditions for the triplex RAA-LFD system. (A) Optimization of the primer and probe concentrations. Lane 1: primers and probes of BCoV, IBRV, and BVDV were 0.150 µM (probe 0.040 µM), 0.200 µM (probe 0.060 µM), and 0.100 µM (probe 0.020 µM), respectively; Lane 2: the primers and probes of BCoV, IBRV, and BVDV were 0.200 µM (probe 0.040 µM), 0.150 µM (probe 0.060 µM), and 0.050 µM (probe 0.020 µM), respectively; Lane 3: the primers and probes of BCoV, IBRV, and BVDV were 0.180 µM (probe 0.040 µM), 0.180 µM (probe 0.050 µM), and 0.050 µM (probe 0.020 µM), respectively; Lane 4: the primers and probes of BCoV, IBRV, and BVDV were 0.135 µM (probe 0.040 µM), 0.135 µM (probe 0.040 µM), and 0.135 µM (probe 0.040 µM), respectively. Lane NC: Negative control. The optimum reaction concentrations of BCoV, IBRV, and BVDV were 0.135 µM (0.040 µM), 0.135 µM (0.040 µM), and 0.135 µM (0.040 µM), respectively. (B) Optimization of the reaction time. Lanes 1–8: reaction times ranging from 11 to 18 min; Lane NC: Negative control. The best reaction time was determined to be 20 min. (C) Optimization of the reaction temperature. Lanes 1–6: the reaction temperature was 37–42°C, Lane NC: negative control. The optimum reaction temperature was determined to be 39°C.

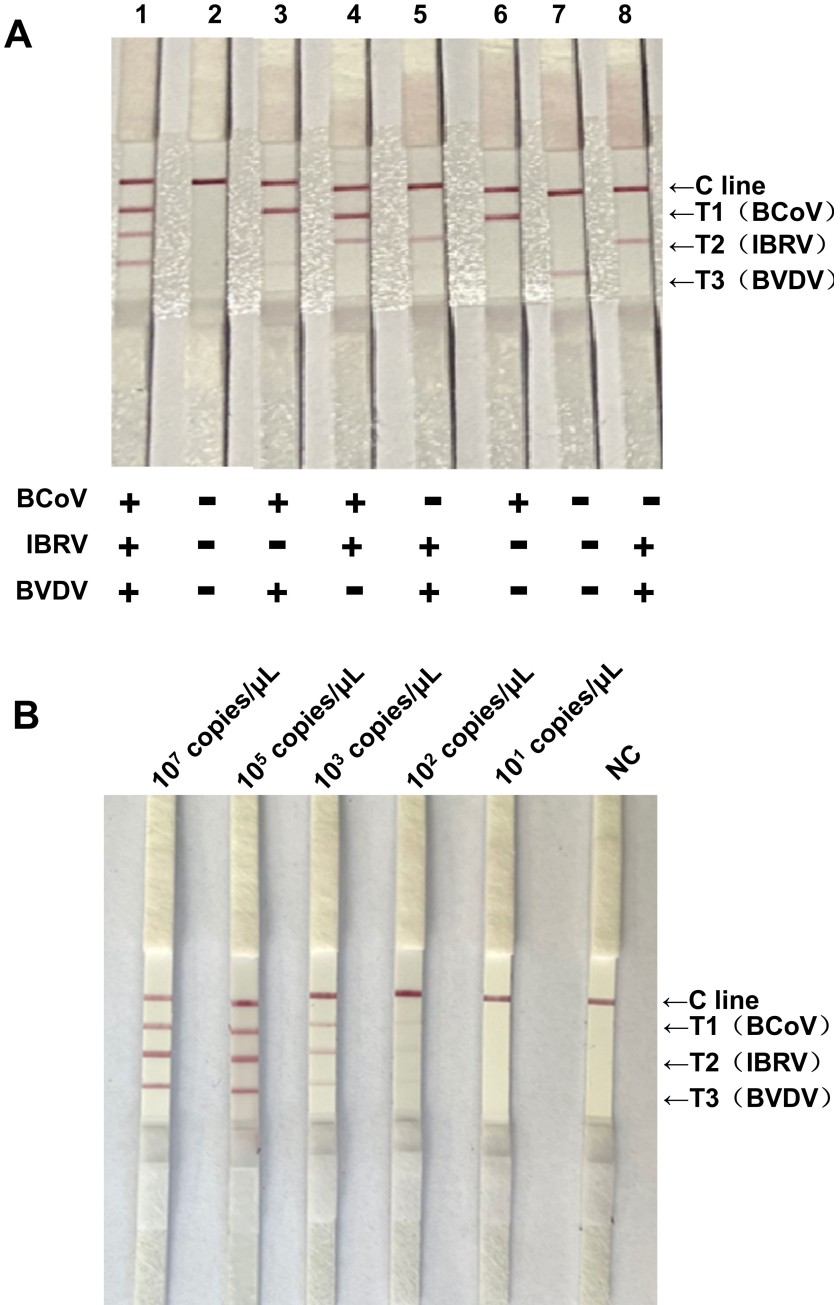

**FIG 6** Specificity and sensitivity of the triplex RAA-LFD system. (A) Specific detection. Lane 1: positive plasmid of BCoV, IBRV, and BVDV; Lane 2: negative control; Lane 3: positive plasmid of BCoV and BVDV; Lane 4: positive plasmid of BCoV and IBRV; Lane 5: positive plasmid of IBRV and BVDV; Lane 6: positive plasmid of BCoV; Lane 7: positive plasmid of BVDV; Lane 8: positive plasmid of IBRV. (B) Sensitivity detection. Lanes 1–5: the template concentrations of BCoV, IBRV, and BVDV ranged from $10^7$ copies/µL to $10^1$ copies/µL; Lane 6: NC, negative control.

## DISCUSSION

As important pathogens affecting the growth and development of the cattle industry, BCoV, IBRV, and BVDV have caused substantial economic losses. Simple, convenient, accurate, rapid, and visual detection technology is essential for the prevention and control of viral diseases. Isothermal amplification technology is an easy, low-cost, and fast method for early monitoring and prevention of infection by pathogens

**TABLE 1** Results of clinical sample detection using triplex RAA-LFD, RT-qPCR, and PCR

| Method | Virus | No. of samples | No. positive | No. negative | Positive rate (%) |
|---|---|---|---|---|---|
| Triplex RAA-LFD | BCoV | 73 | 62 | 11 | 84.93 |
| | IBRV | 73 | 4 | 69 | 5.48 |
| | BVDV | 73 | 12 | 61 | 16.44 |
| RT-qPCR | BCoV | 73 | 63 | 10 | 86.30 |
| | IBRV | 73 | 5 | 68 | 6.85 |
| | BVDV | 73 | 14 | 59 | 19.18 |
| PCR | BCoV | 73 | 26 | 47 | 35.62 |
| | IBRV | 73 | 4 | 69 | 5.48 |
| | BVDV | 73 | 8 | 65 | 10.96 |

(32). RAA represents an advanced isothermal nucleic acid amplification technique in which recombinase-primer complexes facilitate targeted DNA-RNA amplification at constant temperatures (37–42°C), circumventing the thermal cycling requirements of conventional PCR (33). The detection of RAA amplification products can be achieved through three principal methods differentiated by platform requirements: (i) agarose gel electrophoresis (AGE), which offers procedural simplicity but is constrained by instrumentation dependence and potential ethidium bromide contamination hazards (34); (ii) exonuclease III (EXO) probe-based fluorometric analysis, which demonstrates superior sensitivity yet necessiating specialized photometric instrumentation, thereby limiting field applicability (35); and (iii) LFD immunochromatography, which employs biotin-fluorescein conjugate systems with colloidal gold nanoparticles, enables rapid (<5 min) visual interpretation through colorimetric signal development while maintaining operational simplicity (36). In recent years, the RAA-LFD assay has been successfully applied for the detection of various pathogens, such as porcine circovirus type 2 (37), duck circovirus (38), porcine parvovirus (39), influenza A virus and influenza B virus (40). In this study, we developed a triplex RAA-LFD assay for simultaneously detecting viral pathogens (BCoV, IBRV, and BVDV), providing a practical technical solution for the early diagnosis of common viral pathogens in grassroots livestock farms.

In recent years, various techniques for detecting BCV, IBRV, and BVDV have emerged, including conventional PCR, qPCR, and droplet digital PCR (ddPCR). Basanti Brar et al. developed a multiplex PCR method with detection rates of 16.67% for BVDV and 0.9% for BCV (41). Although cost-effective, this PCR-based technique is less sensitive, has more complex procedures, and has a longer detection time than the triplex RAA-LFD assay. Linghao Li et al. developed a multiplex qPCR assay with detection limits as low as 15.1 and 4.99 copies/µL for BVDV and IBRV, respectively (42). Although highly sensitive and specific, this qPCR approach relies heavily on specialized laboratory infrastructure and expensive instruments, requiring 2–3 h for analysis. In contrast, the triplex RAA-LFD technique offers significant advantages in terms of simplicity and rapidity. Furthermore, Junzhen Chen and Zhichao Yu et al. developed a droplet digital PCR (ddPCR) assay with detection limits of 1 copy/µL for BCoV and 4.45 copies/µL for BoHV-1 (3, 43), demonstrating enhanced sensitivity and accuracy. However, this method has a stronger dependence on sophisticated instrumentation and incurs higher costs. In contrast, the triplex RAA-LFD technique requires only a constant-temperature water bath and stable test strips, enabling visual detection within 20 min with simplified operation, making it more suitable for rapid field-based rapid diagnosis. To further validate the efficacy of our triplex RAA-LFD assay, we conducted comparative analyses with other RAA-LFD detection systems. Notably, Yufeng Xiong et al. achieved a detection limit of 10 copies/µL for dengue virus (DENV) within 20 min (23), whereas Wenjing Wang's group reported $10^2$ copies/µL sensitivity for avian infectious laryngotracheitis virus (ILTV) in 23 min (24). Similarly, Li Huihui et al. demonstrated detection limits of $10^3$ copies/µL for both BNov and BRV within 20 min (44). These published RAA-LFD methods are limited to detecting one or two pathogens with sensitivities ranging from $10^1$ to $10^3$ copies/µL. In

**TABLE 2** Sensitivities, specificities, kappa values, coincidence rate, and positive or negative predictive values of the triplex RAA-LFD and RT-qPCR or PCR for the detection of BCoV, IBRV, and BVDV[a]

| | Virus | | RT-qPCR | | | PCR | | |
|---|---|---|---|---|---|---|---|---|
| | | | Pos | Neg | Total | Pos | Neg | Total |
| Triplex RAA-LFD | BCoV | Pos | 62 | 0 | 62 | 26 | 36 | 62 |
| | | Neg | 1 | 10 | 11 | 0 | 11 | 11 |
| | | Total | 63 | 10 | 73 | 26 | 47 | 73 |
| | | Sen: | Spe: | K: 0.94 | | Sen: | Spe: | K: 0.33 |
| | | 98.41% | 100.00% | | | 100.00% | 23.40% | |
| | | PPV: | NPV: | CR: | | PPV: | NPV: | CR: |
| | | 100.00% | 90.91% | 98.63% | | 41.94% | 100.00% | 50.68% |
| Triplex RAA-LFD | IBRV | Pos | 4 | 0 | 4 | 3 | 1 | 4 |
| | | Neg | 1 | 68 | 69 | 1 | 68 | 69 |
| | | Total | 5 | 68 | 73 | 4 | 69 | 73 |
| | | Sen: | Spe: | K:0.88 | | Sen: | Spe: | K: 0.73 |
| | | 80.00% | 100.00% | | | 75.00% | 98.55% | |
| | | PPV: | NPV: | CR: | | PPV: | NPV: | CR: |
| | | 100.00% | 93.15% | 98.63% | | 75.00% | 98.55% | 97.26% |
| Triplex RAA-LFD | BVDV | Pos | 12 | 0 | 12 | 7 | 5 | 12 |
| | | Neg | 2 | 59 | 61 | 1 | 60 | 61 |
| | | Total | 14 | 59 | 73 | 8 | 65 | 73 |
| | | Sen: | Spe: | K: 0.91 | | Sen: | Spe: | K: 0.65 |
| | | 85.71% | 100.00% | | | 87.50% | 92.31% | |
| | | PPV: | NPV: | CR: | | PPV: | NPV: | CR: |
| | | 100.00% | 96.72% | 97.26% | | 58.33% | 98.36% | 91.78% |

[a]Pos, positive; Neg, negative; Sen, sensitivity; Spe, specificity; K, Kappa value; CR, coincidence rate; PPV, positive predictive value; NPV, negative predictive value.

contrast, our optimized triplex RAA-LFD platform enables simultaneous visual detection of three pathogens (BCV, IBRV, and BVDV) with sensitivities of $4.13 \times 10^3$, $3.20 \times 10^3$, and $2.21 \times 10^3$ copies/µL respectively, while maintaining a consistent 20 min detection time. This comparative evaluation demonstrated that our multiplex system preserves satisfactory sensitivity while significantly improving detection throughput for multiple pathogens. Furthermore, clinical sample testing demonstrated that the triplex RAA-LFD assay, although slightly less sensitive than qPCR, exhibited significantly higher detection sensitivity than conventional PCR while enabling simultaneous visual detection of three pathogens with acceptable accuracy. This cost-effective approach makes it suitable for rapid pathogen screening in field settings. However, the reliance on commercial nucleic acid extraction kits increases costs, necessitating future research into simpler, low-cost alternatives (e.g., thermal lysis). Additionally, field applications require further optimization of the assay's robustness against environmental interference and enhanced specificity to improve its reliability for field-based diagnostics.

In conclusion, this study represents the first development of a multiplex RAA-LFD assay for the simultaneous detection of BCV, IBRV, and BVDV. The established method has notable advantages in speed, cost-effectiveness, and operational simplicity and has strong potential as a practical tool for field-based screening of common pathogens in neonatal calves.

## Conclusion

The present study successfully established the triplex RAA-LFD technique to detect BCoV, IBRV, and BVDV in one test simultaneously. This method has the advantages of high sensitivity, strong specificity, and short analysis time. Therefore, it is well-suited for onsite BCoV, IBRV, and BVDV diagnoses and molecular epidemiological surveys, especially in resource-limited or field settings.

## ACKNOWLEDGMENTS

Yingcai Ma, Data curation, Investigation, Writing-original draft, Writing-review and editing | Xinhao Wang, Investigation, Writing-review and editing | Rulong Chen, Investigation, Supervision, Writing-review and editing | Keer Dong, Conceptualization, Data curation | Xiaojie Yu, Conceptualization, Data curation | Na Li, Supervision, Writing-review and editing | Li Yang, Supervision, Writing-review and editing | Qi Zhong, Supervision, Writing-review and editing | Gang Yao, Supervision, Writing-review and editing | Xuelian Ma, Funding acquisition, Supervision, Writing-original draft, Writing-review and editing. All authors have read and agreed to the published version of the manuscript.

This work was supported by the Key Research and Development Program of Xinjiang Uygur Autonomous Region (Grant No. 2025B02018), the Outstanding Youth Science Foundation Project of the Natural Science Foundation of Xinjiang Uygur Autonomous Region (Grant No. 2024D01E05), the Autonomous Region Key Research and Development Program Project (Grant No. 2024B02011-2), and the National Beef and Yak Industry Technology System (CARS-37). The funding bodies did not contribute to the conception or design of the study, the analysis or interpretation of the data, or the writing of the manuscript.

## AUTHOR AFFILIATIONS

[1]College of Veterinary Medicine, Xinjiang Agricultural University, Urumqi, China
[2]Xinjiang Key Laboratory of New Drug Research and Development for Herbivores, Urumqi, China
[3]Xinjiang Tianlai Breeding Co., Ltd, Bole, China
[4]Institute of Animal Science, Xinjiang Academy of Animal Sciences, Urumqi, China

## AUTHOR ORCIDs

Yingcai Ma ⓘ http://orcid.org/0009-0008-0314-2393
Xuelian Ma ⓘ http://orcid.org/0000-0001-7336-2211

## FUNDING

| Funder | Grant(s) | Author(s) |
|---|---|---|
| The Key Research and Development Program of Xinjiang Uygur Autonomous Region | 2025B02018 | Xuelian Ma |
| The Outstanding Youth Science Foundation Project of Xinjiang Uygur Autonomous Region | 2024D01E05 | Xuelian Ma |
| The Autonomous Region's Key Research and Development Program Project | 2024B02011-2 | Na Li |
| The National Beef and Yak Industry Technology System | CARS-37 | Rulong Chen |

## AUTHOR CONTRIBUTIONS

Yingcai Ma, Data curation, Investigation, Writing – original draft, Writing – review and editing | Xinhao Wang, Investigation, Writing – review and editing | Rulong Chen, Investigation, Supervision, Writing – review and editing | Keer Dong, Conceptualization, Data curation | Xiaojie Yu, Conceptualization, Data curation | Na Li, Supervision, Writing – review and editing | Li Yang, Supervision, Writing – review and editing | Qi Zhong, Supervision, Writing – review and editing | Gang Yao, Supervision, Writing – review and editing | Xuelian Ma, Funding acquisition, Supervision, Writing – original draft, Writing – review and editing

## DATA AVAILABILITY

The data are contained within the article.

## ADDITIONAL FILES

The following material is available online.

### Supplemental Material

**Supplemental material (Spectrum01628-25-s0001.pdf).** Fig. S1 and S2; Supplemental table legends.
**Table S1 (Spectrum01628-25-s0002.xlsx).** RAA assay primers of BCoV, IBRV, and BVDV.
**Table S2 (Spectrum01628-25-s0003.xlsx).** Primers of PCR, RT-qPCR, and probes.
**Table S3 (Spectrum01628-25-s0004.xlsx).** The optimum primers and probes of RAA-LFD.
**Table S4 (Spectrum01628-25-s0005.xlsx).** The sample test statistics.

### Open Peer Review

**PEER REVIEW HISTORY (review-history.pdf).** An accounting of the reviewer comments and feedback.

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
