## [Reviewer comments · Microbiology Spectrum]

Microbiology Spectrum

Developing a Recombinase-aided Amplification Method Combined with a Lateral Flow Dipstick Assay for Rapid Triplex Detection of Bovine Coronavirus, Infectious Bovine Rhinotracheitis Virus, and Bovine Viral Diarrhea Virus

Yingcai Ma, Xinhao Wang, Rulong Chen, Keer Dong, Xiaojie Yu, Na Li, Li Yang, Qi Zhong, Gang Yao, and Xuelian Ma

Corresponding Author(s): Xuelian Ma, Xinjiang Agricultural University

Review Timeline:

Submission Date:	May 26, 2025
Editorial Decision:	August 7, 2025
Revision Received:	October 13, 2025
Accepted:	October 27, 2025

Editor: Alexander Bello

Reviewer(s): The reviewers have opted to remain anonymous.

Transaction Report:

DOI: <https://doi.org/10.1128/spectrum.01628-25>

Re: Spectrum01628-25 (Developing a Recombinase-aided Amplification Method Combined with a Lateral Flow Dipstick Assay for Rapid Triplex Detection of Bovine Coronavirus, Infectious Bovine Rhinotracheitis Virus, and Bovine Viral Diarrhea Virus)

Dear Dr. Xuelian Ma:

Thank you for the privilege of reviewing your work. Below you will find my comments, instructions from the Spectrum editorial office, and the reviewer comments.

Please seek help to improve the English grammar in the manuscript.

Revision Guidelines

Sincerely,
Alexander Bello
Editor
Microbiology Spectrum

Reviewer #2 (Comments for the Author):

The authors develop a recombinase-aided amplification assay that used a lateral flow assays to detect 3 bovine viral diseases.

The sensitivity was determined for each of the 3 viruses in the range of 1000 copies. The assay was evaluated with field

samples from cattle with clinical disease. The limit of detection of the real-time PCRs should be much lower, can the authors compare the limit of detection with that of the PCR assays.

Minor issues:

line 69 how is this test field deployable when extraction of genetic material is required. The authors do not mention the use of an extraction method that can be used in the field.

Line 248 The details of the detection are not clear. The details of the extraction step is required for the RT-RAA-LFD test.

How is this test able to be used as a point-of-care test? The test requires extraction, how would this be done in the field. Also the test requires an amplification step at 39{degree sign}C, how would this be performed in the field. Then the test requires reading on a LFD. How is the assay more easily performed compared to real-time PCR. If there are no real-time PCR machines then it could be used, however, the test as described is not really a point-of-care test and has similar issues in performing compared to real-time PCR.

Reviewer #3 (Comments for the Author):

Although this study describes a very useful, rapid and cheaper multiplex diagnostic method for detecting three common viral diseases of cattle, the write-up needs to be well articulated. Some methods are not clearly described and specifically, there is a need to use a known negative sample for the specificity test as well.

Comments for the Authors:

The manuscript by Yingcai Ma et al. describes the successful development of a triplex RAA-LFD technique for the simultaneous detection of BCoV, IBRV, and BVDV in a single test. The method offers advantages such as high sensitivity, strong specificity, and rapid analysis, making it highly suitable for field deployment in multiplex diagnosis of common cattle diseases and for epidemiological surveys. The authors collected and tested samples from diarrheic cattle on commercial farms. Therefore, this study is valuable and well-suited for the onsite diagnosis of BCoV, IBRV, and BVDV, as well as molecular epidemiological surveys, particularly in resource-limited regions. However, a more detailed presentation of the methods, better organization of results following the sequence of the materials and methods, and a comprehensive rewrite of the discussion are needed. The discussion requires a thorough revision to address this study's findings, related research, and the method's limitations. Below, I have highlighted this and other areas where I believe the manuscript could be improved. Additionally, Assistance from an English editor is also recommended.

Abstract:

Line 49: Replace “anal” with “rectal”

Importance:

- Lines 61-62: I will remove the full meanings of the abbreviated viruses since that has been done above.
- Line 63: The “specificity”test should also include true negative
- Line 68: No need to include “due to methodological differences”.
- Line 77: Remove “all of the scale
- Line 79: Change “Increasing challenges” to “increasingly challenging”
- Line 90: Please take a closer look at “conjunctivitis ???”. It is the inflammation of the conjunctiva.
- Line 93-97:It is a very long sentence. Kindly rewrite.
- Line 97-98: You could recast this sense this way “ These viral infections in cattle cause significant financial losses for the beef industry”.
- Line 99-100: The sentence is hanging as the problem is that of prevention and control , but what the authors offer is rapid, accurate, and cheap diagnosis. What would have thought that it is the diagnostic challenges that are stated, which will of course affect control and prevention. Please restate the problem.
- Line 100. Early detection and early diagnosis are the same. Please choose one.
- Line 104: PCR in full not needed. It is conventional. Kindly remove.
- Line 105: RT-qPCR. The full meaning is, Reverse transcriptase quantitative PCR.
- Line 122-123: Change “were” to “are”
- Line 124: use RAA-LFD abbreviation
- Line 127: in the field to detect parasites (21), viruses(22-24) & other pathogens (20).

Materials and Methods

-

-
- Line 139- 142 : There is need for the reference of the source of the plasmid
- Line 143: The viruses cited BCoV etc. needs to be referenced.
- Line 144: competitive positive controls the assay
- Line 145: Kindly specify; are they pooled rectal and nasal samples or individual swab samples. Change anal to rectal swab. That is ; 73 rectal and nasal swabs: Are they pooled together or 73 rectal and 73 nasal swabs separately? Also Specify the condition of sample collection and preservation. Date of collection could be added.
- Line 148: Write “until testing”
- Line 149: Add references or specify the source of nucleic acid extraction.
- Line 150: viral RNA of BCoV and BVDV sentence is not clear whether from sample or plasmid
- Line 150 - 160: Needs to be clearer; recast
- Line 162: remove quality
- Line 163-168: The details of the plasmids are not clear. .e.g how do they clone and insert gB genes? The method is unclear
- Line 162-176- recorrect
- Line 177: preparations of LFD
- Line 178: move “test” after “LFD”
- Line 179: What is NC membrane? Kindly write in full meaning first. I presume it is negative control.
- Line 225, remove “040 μM),”
- Line 232: remove “quality and plasmids” Simply write positive control in the the whole document. You can specify if nucleic acid only ore the entire plamid.
- Line 233: 1 nl; recombine IBRV
- Line 239: Write the positive controls
- Line 246: remove quality
- Line 236: The specificity is missing
- Line 245: title should be changed
- Line 246: What is RT? Kindly write it in full as it has not been mentioned elsewhere
- Line 247: should be recast to rectal swab instead of anal swab
- Line 246-248: Recast the sentence to te detection of BcOv, IBRV, and BVDV was performed using the triple RAA- LFO assay on 73 swab samples collected diareheric bovine specify whether rectal swab or nasal swab or both were used for the test
- Line 251:

Results

- Line 260-284: should be removed and moved to method as it is not a result, it is a repetition
- Line 281; Recast, Using the recombinant gene for each target virus as template
- Line 285: screening of the optimal primers should be moved to method
- Line 286- 289: The paragraph is more of a method(move to method)

- Line 290: Specify R3 and F3
- Line 311: After BCDV, add respectively....
- Line 313: Replace positive quality contol plasmids with positive control recombinant gene
- Line 317-321: Recast (susceptible pathogen)
- Line 309- 317: Move upwards to a newly created paragraph
- Line 328i : What quantity and concentration are tamara and digoxin added. Should move to Mand M; with more clarity
- Line 362- 366: should be removed
- Line 367:The title should be sensitivity only instead of specificity and sensitivity
- Line 370: The title should be changed; ..RRA using clinical samples
- Line 378-380: should be removed

Discussion

- Line 394- 410 : Recast and send to introduction
- Line 411-415: Introduce your overall result finding and discuss I therefore suggest you recast this whole paragraph
- Lin3 415- 448 Discussion needs to be rewritten taking into account your results and findings
- Line 463: add 'the' after analysis

Comments for the Authors:

The manuscript by Yingcai Ma et al. describes the successful development of a triplex RAA-LFD technique for the simultaneous detection of BCoV, IBRV, and BVDV in a single test. The method offers advantages such as high sensitivity, strong specificity, and rapid analysis, making it highly suitable for field deployment in multiplex diagnosis of common cattle diseases and for epidemiological surveys. The authors collected and tested samples from diarrheic cattle on commercial farms. Therefore, this study is valuable and well-suited for the onsite diagnosis of BCoV, IBRV, and BVDV, as well as molecular epidemiological surveys, particularly in resource-limited regions. However, a more detailed presentation of the methods, better organization of results following the sequence of the materials and methods, and a comprehensive rewrite of the discussion are needed. The discussion requires a thorough revision to address this study's findings, related research, and the method's limitations. Below, I have highlighted this and other areas where I believe the manuscript could be improved. Additionally, Assistance from an English editor is also recommended.

Abstract:

Line 49: Replace “anal” with “rectal”

Importance:

- Lines 61-62: I will remove the full meanings of the abbreviated viruses since that has been done above.
- Line 63: The “specificity” test should also include true negatives
- Line 68: No need to include “due to methodological differences”.
- Line 77: Remove “all of the scale
- Line 79: Change “Increasing challenges” to “increasingly challenging”
- Line 90: Please take a closer look at “conjunctivitis ???”. It is the inflammation of the conjunctiva.
- Line 93-97: It is a very long sentence. Kindly rewrite.
- Line 97-98: You could recast this sense this way “ These viral infections in cattle cause significant financial losses for the beef industry”.
- Line 99-100: The sentence is hanging as the problem is that of prevention and control, but what the authors offer is rapid, accurate, and cheap diagnosis. What would have thought that it is the diagnostic challenges that are stated, which will of course affect control and prevention. Please restate the problem.
- Line 100. Early detection and early diagnosis are the same. Please choose one.
- Line 104: PCR in full not needed. It is conventional. Kindly remove.
- Line 105: RT-qPCR. The full meaning is, Reverse transcriptase quantitative PCR.
- Line 122-123: Change “were” to “are”
- Line 124: use RAA-LFD abbreviation
- Line 127: in the field to detect parasites (21), viruses(22-24) & other pathogens (20).

Materials and Methods

-

-
- Line 139- 142 : There is need for the reference of the source of the plasmid
- Line 143: The viruses cited BCoV etc. needs to be referenced.
- Line 144: competitive positive controls the assay
- Line 145: Kindly specify; are they pooled rectal and nasal samples or individual swab samples. Change anal to rectal swab. That is ; 73 rectal and nasal swabs: Are they pooled together or 73 rectal and 73 nasal swabs separately? Also Specify the condition of sample collection and preservation. Date of collection could be added.
- Line 148: Write “until testing”
- Line 149: Add references or specify the source of nucleic acid extraction.
- Line 150: viral RNA of BCoV and BVDV sentence is not clear whether from sample or plasmid
- Line 150 - 160: Needs to be clearer; recast
- Line 162: remove quality
- Line 163-168: The details of the plasmids are not clear. .e.g how do they clone and insert gB genes? The method is unclear
- Line 162-176- recorrect
- Line 177: preparations of LFD
- Line 178: move “test” after “LFD”
- Line 179: What is NC membrane? Kindly write in full meaning first. I presume it is negative control.
- Line 225, remove “040 μM),”
- Line 232: remove “quality and plasmids” Simply write positive control in the the whole document. You can specify if nucleic acid only ore the entire plamid.
- Line 233: 1 nl; recombine IBRV
- Line 239: Write the positive controls
- Line 246: remove quality
- Line 236: The specificity is missing
- Line 245: title should be changed
- Line 246: What is RT? Kindly write it in full as it has not been mentioned elsewhere
- Line 247: should be recast to rectal swab instead of anal swab
- Line 246-248: Recast the sentence to te detection of BcOv, IBRV, and BVDV was performed using the triple RAA- LFO assay on 73 swab samples collected diareheric bovine specify whether rectal swab or nasal swab or both were used for the test
- Line 251:

Results

- Line 260-284: should be removed and moved to method as it is not a result, it is a repetition
- Line 281; Recast, Using the recombinant gene for each target virus as template
- Line 285: screening of the optimal primers should be moved to method
- Line 286- 289: The paragraph is more of a method(move to method)

- Line 290: Specify R3 and F3
- Line 311: After BCDV, add respectively....
- Line 313: Replace positive quality control plasmids with positive control recombinant gene
- Line 317-321: Recast (susceptible pathogen)
- Line 309- 317: Move upwards to a newly created paragraph
- Line 328i : What quantity and concentration are tamara and digoxin added. Should move to Mand M; with more clarity
- Line 362- 366: should be removed
- Line 367: The title should be sensitivity only instead of specificity and sensitivity
- Line 370: The title should be changed; ..RRA using clinical samples
- Line 378-380: should be removed

Discussion

- Line 394- 410: Recast and send to introduction
- Line 411-415: Introduce your overall result finding and discuss I therefore suggest you recast this whole paragraph
- Lin3 415- 448 Discussion needs to be rewritten taking into account your results and findings
- Line 463: add 'the' after analysis

Dear Editors and Reviewers,

Thank you for your letter and for the reviewers' comments concerning our manuscript entitled "Developing a Recombinase-aided Amplification Method Combined with a Lateral Flow Dipstick Assay for Rapid Triplex Detection of Bovine Coronavirus, Infectious Bovine Rhinotracheitis Virus, and Bovine Viral Diarrhea Virus" (ID: Spectrum01628-25). These comments are all valuable and very helpful for revising and improving our paper, as well as important for guiding our research. We have studied the comments carefully and have made corrections that we hope will meet with approval. The revised portions are highlighted in yellow in the manuscript. The main responses to the reviewer comments are as follows.

Editor:

(1) Please seek help to improve the English grammar in the manuscript.

Response: Thank you for your reminder. We have had the manuscript thoroughly proofread for spelling and grammar errors by professional polishing agencies and have made corresponding revisions in the manuscript.

Response: Thank you for your reminder. We look forward to having our research findings published in your esteemed journal. We sincerely appreciate the valuable comments and suggestions provided by the editors and reviewers, as they serve as important avenues for us to gain new perspectives. These insights will inspire new ideas for our scientific research endeavors and facilitate the steady advancement and refinement of our ongoing research work.

Reviewer 2 (Comments for the Author):

The authors develop a recombinase-aided amplification assay that used a lateral flow assays to detect 3 bovine viral diseases.

(1) The sensitivity was determined for each of the 3 viruses in the range of 1000 copies. The assay was evaluated with field samples from cattle with clinical disease. The limit of detection of the real-time PCRs should be much lower, can the authors compare the limit of detection with that of the PCR assays.

Response: Thank you for your suggestion. After testing the clinical samples via three detection techniques (the triplex RAA-LFD assay, qPCR, and PCR), we conducted a detailed comparison of the test results, including comparisons between the triplex RAA-LFD assay and the qPCR and PCR methods. The specific test results and analysis are detailed in Table 2.

(2) line 69 how is this test field deployable when extraction of genetic material is required. The authors do not mention the use of an extraction method that can be used in the field.

Response: Thank you for your suggestions. The triplex RAA-LFD assay we developed primarily targets grassroots farms. During our research, we learned that most farms are equipped with simple operational laboratories capable of nucleic acid extraction. Therefore, we adopted nucleic acid extraction kits widely accepted by farms to perform nucleic acid extraction for subsequent experiments, which achieved favorable results. For farms without laboratory facilities, we employed a thermal lysis method for crude nucleic acid extraction and used the obtained nucleic acids for subsequent experiments, which also enabled visual detection. However, the detection performance of the thermal lysis method was not as effective as that of the kit-based extraction method. For the specific experimental procedure, we have carefully revised the Methods section, supplementing specific details from nucleic acid extraction to sample detection. The relevant content has been incorporated into the manuscript.

(3) Line 248 The details of the detection are not clear. The details of the extraction step is required for the RT-RAA-LFD test.

Response: Thank you for your suggestions. We have carefully revised the Methods section, supplementing specific details from nucleic acid extraction to sample detection. The relevant content has been incorporated into the manuscript.

(4) How is this test able to be used as a point-of-care test? The test requires extraction, how would this be done in the field. Also the test requires an amplification step at 39{degree sign}C, how would this be performed in the field. Then the test requires reading on a LFD. How is the assay more easily performed compared to real-time PCR. If there are no real-time PCR machines then it could be used, however, the test as described is not really a point-of-care test and has similar issues in performing compared to real-time PCR.

Response: Thank you for your valuable suggestions. The triplex RAA-LFD assay we established is not comparable in difficulty to qPCR. First, our RAA-LFD method only requires isothermal amplification, which can be directly performed in a 39°C

constant-temperature water bath, eliminating the need for expensive qPCR instruments or specialized *Homo sapiens* operators. Second, in terms of response time, it takes only 15 minutes in a 39°C water bath, followed by the application of the reaction droplets onto the LFD, and the results can be visually observed within 5 minutes, indicating a significantly shorter response time. Furthermore, regarding detection efficiency, the triplex RAA-LFD method enables simultaneous monitoring of three pathogens in a single reaction, whereas qPCR suffers from reduced efficiency, prolonged duration, and increased costs and requires specialized *Homo sapiens* operators when performing multiplex detection. Therefore, our established triplex RAA-LFD detection method is more practical for grassroots livestock farms. We have meticulously revised the methodology section, supplementing the detailed procedures from nucleic acid extraction to sample detection.

Reviewer 3 (Comments for the Author):

(1) Although this study describes a very useful, rapid and cheaper multiplex diagnostic method for detecting three common viral diseases of cattle, the write-up needs to be well articulated. Some methods are not clearly described and specifically, there is a need to use a known negative sample for the specificity test as well.

Response: We sincerely appreciate this valuable suggestion. We have meticulously revised the description of the methodology section in the paper and supplemented specific details from nucleic acid extraction to sample detection. In the specificity testing section, we conducted tests on both known positive and negative samples for BCoV, IBRV, and BVDV (lanes 1 and 2 in Figure 6A), which demonstrated good specificity. Additionally, the collected samples were subjected to RAA-LFD, qPCR, and PCR, with 10 samples testing negative by all three methods. The detailed test results are summarized in Supplementary Table S4.

(2) The manuscript by Yingcai Ma et al. describes the successful development of a triplex RAA-LFD technique for the simultaneous detection of BCoV, IBRV, and BVDV in a single test. The method offers advantages such as high sensitivity, strong specificity, and rapid analysis, making it highly suitable for field deployment in multiplex diagnosis of common cattle diseases and for epidemiological surveys. The authors collected and tested samples from diarrheic cattle on commercial farms. Therefore, this study is valuable and well-suited for the onsite diagnosis of BCoV, IBRV, and BVDV, as well as molecular epidemiological surveys, particularly in resourcelimited regions. However, a more detailed presentation of the

methods, better organization of results following the sequence of the materials and methods, and a comprehensive rewrite of the discussion are needed. The discussion requires a thorough revision to address this study's findings, related research, and the method's limitations. Below, I have highlighted this and other areas where I believe the manuscript could be improved. Additionally, Assistance from an English editor is also recommended.

Response: Thank you for your valuable suggestions. We have meticulously revised the description and methodology sections of the paper, supplemented specific details from nucleic acid extraction to sample detection, streamlined the results section, and comprehensively rewritten the discussion chapter to cover the findings of this study, related research, and the limitations of the method. Additionally, we invited a professional English editor to review the article thoroughly to ensure compliance with the English writing requirements. The specific modifications are highlighted in yellow within the manuscript.

Abstract:

Line 49: Replace "anal" with "rectal"

Response: Thank you for your reminder. We have replaced "anal" with "rectal" and revised the manuscript accordingly.

Importance:

- Lines 61-62: I will remove the full meanings of the abbreviated viruses since that has been done above.

Response: Thank you for your reminder. We have corrected the relevant errors in the manuscript.

- Line 63: The "specificity" test should also include true negatives

Response: Response: Thank you for your suggestions. We have meticulously revised the description of the methodology section in the paper and supplemented specific details from nucleic acid extraction to sample detection. In the specificity testing section, we conducted tests on both known positive and negative samples for BCoV, IBRV, and BVDV (lanes 1 and 2 in Figure 6A), which demonstrated good specificity. Additionally, the collected samples were subjected to RAA-LFD, qPCR, and PCR, with 10 samples testing negative by all three methods. The detailed test results are summarized in Supplementary Table S4.

- Line 68: No need to include "due to methodological differences".

Response: Thank you for your reminder. We have revised the manuscript accordingly.

- Line 77: Remove "all of the scale"

Response: Thank you for your reminder. We have revised the manuscript accordingly.

- Line 79: Change "Increasing challenges" to "increasingly challenging"

Response: Thank you for your reminder. We have revised the manuscript accordingly.

● Line 90: Please take a closer look at "conjunctivitis ???". It is the inflammation of the conjunctiva.

Response: Thank you for your suggestion. We have verified the language expression. IBRV infection in calves can indeed cause respiratory, conjunctivitis, meningitis, and reproductive tract clinical symptoms. The relevant content has been revised in the manuscript.

- Line 93-97: It is a very long sentence. Kindly rewrite.

Response: Thank you for your suggestions. We have revised the language accordingly, and the relevant content has been updated in the manuscript.

● Line 97-98: You could recast this sense this way "These viral infections in cattle cause significant financial losses for the beef industry".

Response: Thank you for your suggestions. We have revised the language accordingly, and the relevant content has been updated in the manuscript.

● Line 99-100: The sentence is hanging as the problem is that of prevention and control, but what the authors offer is rapid, accurate, and cheap diagnosis. What would have thought that it is the diagnostic challenges that are stated, which will of course affect control and prevention. Please restate the problem.

Response: Thank you for your suggestions. We have revised the language accordingly, and the relevant content has been updated in the manuscript.

- Line 100. Early detection and early diagnosis are the same. Please choose one.

Response: Thank you for your suggestions. We have revised the language accordingly, and the relevant content has been updated in the manuscript.

- Line 104: PCR in full not needed. It is conventional. Kindly remove.

Response: Thank you for your suggestions. We have revised the language accordingly, and the relevant content has been updated in the manuscript.

- Line 105: RT-qPCR. The full meaning is, Reverse transcriptase quantitative PCR.

Response: Thank you for your suggestions. We have revised the language accordingly, and the relevant content has been updated in the manuscript.

- Line 122-123: Change "were" to "are"

Response: Thank you for your suggestions. We have revised the language accordingly, and the relevant content has been updated in the manuscript.

- Line 124: use RAA-LFD abbreviation

Response: Thank you for your suggestions. We have revised the language accordingly, and the relevant content has been updated in the manuscript.

- Line 127: in the field to detect parasites (21), viruses(22-24) & other pathogens (20).

Response: Thank you for your suggestions. We have revised the language accordingly, and the relevant content has been updated in the manuscript.

Materials and Methods

- Line 139- 142 : There is need for the reference of the source of the plasmid

Response: Thank you for your suggestions. We have revised the language accordingly, and the relevant content has been updated in the manuscript.

- Line 143: The viruses cited BCoV etc. needs to be referenced.

Response: Thank you for your suggestions. We have revised the language accordingly, and the relevant content has been updated in the manuscript.

- Line 144: competitive positive controls the assay

Response: Thank you for your suggestions. We have revised the language accordingly, and the relevant content has been updated in the manuscript.

● Line 145: Kindly specify; are they pooled rectal and nasal samples or individual swab samples. Change anal to rectal swab. That is ; 73 rectal and nasal swabs: Are they pooled together or 73 rectal and 73 nasal swabs separately? Also Specify the condition of sample collection and preservation. Date of collection could be added.

Response: Thank you for your suggestions. From December 2021 to December 2022, 73 rectal and nasal swabs (50 rectal swabs and 23 nasal swabs) were obtained from diarrhoeic cows on commercial bovine farms in Kashgar, Yili and Changji in Xinjiang Province, China. We have revised the language accordingly, and the relevant content has been updated in the manuscript.

- Line 148: Write "until testing"

Response: Thank you for your suggestions. We have revised the language accordingly, and the relevant content has been updated in the manuscript.

- Line 149: Add references or specify the source of nucleic acid extraction.

Response: Thank you for your suggestions. We have revised the language accordingly, and the relevant content has been updated in the manuscript.

- Line 150: viral RNA of BCoV and BVDV sentence is not clear whether from sample or plasmid

Response: Thank you for your suggestions. We have revised the language accordingly, and the relevant content has been updated in the manuscript.

- Line 150 - 160: Needs to be clearer; recast

Response: Thank you for your suggestions. We have revised the nucleic acid extraction section for BCoV, BVDV, and IBRV, and the relevant content has been updated in the original text.

- Line 162: remove quality

Response: Thank you for your suggestions. We have revised the language accordingly, and the relevant content has been updated in the manuscript.

- Line 163-168: The details of the plasmids are not clear. .e.g how do they clone and insert gB genes? The method is unclear

Response: Thank you for your suggestions. We have revised the language accordingly, and the relevant content has been updated in the manuscript.

- Line 162-176- recorrect

Response: Thank you for your suggestions. We have revised the language accordingly, and the relevant content has been updated in the manuscript.

- Line 177: preparations of LFD

Response: Thank you for your suggestions. We have revised the language accordingly, and the relevant content has been updated in the manuscript.

- Line 178: move "test" after "LFD"

Response: Thank you for your suggestions. We have revised the language accordingly, and the relevant content has been updated in the manuscript.

- Line 179: What is NC membrane? Kindly write in full meaning first. I presume it is negative control.

Response: Thank you for your suggestions. We have revised the expression of the NC membrane. The name of the NC membrane is the nitrocellulose (NC) membrane. The relevant content has been revised in the manuscript.

- Line 225, remove "040 μ M),"

Response: Thank you for your reminder. We have deleted "040 μ M)," and the relevant content has been revised in the manuscript.

- Line 232: remove "quality and plasmids" Simply write positive control in the the whole document. You can specify if nucleic acid only ore the entire plamid.

Response: Thank you for your suggestions. We have revised the language accordingly, and the relevant content has been updated in the manuscript.

- Line 233: 1 nl; recombine IBRV

Response: Thank you for your suggestions. We have revised the language accordingly, and the relevant content has been updated in the manuscript.

- Line 239: Write the positive controls

Response: Thank you for your suggestions. We have revised the language accordingly, and the relevant content has been updated in the manuscript.

- Line 246: remove quality

Response: Thank you for your suggestions. We have revised the language accordingly, and the relevant content has been updated in the manuscript.

- Line 236: The specificity is missing

Response: Response: Thank you for your suggestions. We have meticulously revised the description of the methodology section in the manuscript and supplemented specific details from nucleic acid extraction to sample detection. In the specificity testing section, we conducted tests on both known positive and negative samples for BCoV, IBRV, and BVDV (lanes 1 and 2 in Figure 6A), which demonstrated good specificity.

- Line 245: title should be changed

Response: Thank you for your suggestions. We have revised the language accordingly, and the relevant content has been updated in the manuscript.

- Line 246: What is RT? Kindly write it in full as it has not been mentioned elsewhere

Response: Thank you for your suggestions. We have revised the language accordingly, and the relevant content has been updated in the manuscript.

- Line 247: should be recast to rectal swab instead of anal swab

Response: Thank you for your suggestions. We have revised the language accordingly, and the relevant content has been updated in the manuscript.

● Line 246-248: Recast the sentence to the detection of BCoV, IBRV, and BVDV was performed using the triple RAA- LFO assay on 73 swab samples collected diareheric bovine specify whether rectal swab or nasal swab or both were used for the test

Response: Thank you for your suggestions. We have revised the language accordingly, and the relevant content has been updated in the manuscript.

Results

- Line 260-284: should be removed and moved to method as it is not a result, it is a

Repetition

Response: Thank you for your suggestion. The "Triplex RAA-LFD assay strategy" section provides a detailed explanation of the principles underlying the establishment of the triple detection method, with a particular emphasis on the response principle. This does not overlap with the methods section. We believe that placing this content in the first part of the results section can serve as a guiding framework for the entire manuscript, thereby enhancing the logical clarity of the manuscript.

- Line 281; Recast, Using the recombinant gene for each target virus as template

Response: Thank you for your suggestions. We have revised the language accordingly, and the relevant content has been updated in the manuscript.

- Line 285: screening of the optimal primers should be moved to method

Response: Thank you for your suggestion. The section "Screening of the optimal primers for the RAA-LFD assay" presents the results obtained from the experimental process and highlights the research foundation for subsequent experiments. This does not overlap with the methods section. We believe that placing this content in the second part of the results will clarify the manuscript's logic.

- Line 286- 289: The paragraph is more of a method(move to method)

Response: Thank you for your suggestion. We have included this content in the Methods section.

- Line 290: Specify R3 and F3

Response: Thank you for your suggestions. We have revised the language accordingly, and the relevant content has been updated in the manuscript.

- Line 311: After BCDV, add respectively....

Response: Thank you for your suggestions. We have revised the language accordingly, and the relevant content has been updated in the manuscript.

- Line 313: Replace positive quality contol plasmids with positive control recombinant Gene

Response: Thank you for your suggestions. We have revised the language accordingly, and the relevant content has been updated in the manuscript.

- Line 317-321: Recast (susceptible pathogen)

Response: Thank you for your suggestions. We have revised the language accordingly, and the relevant content has been updated in the manuscript.

- Line 309- 317: Move upwards to a newly created paragraph

Response: Thank you for your suggestions. We have revised the language accordingly, and the relevant content has been updated in the manuscript.

- Line 328 : What quantity and concentration are tamara and digoxin added. Should move to Mand M; with more clarity

Response: Thank you for your suggestion. We apologize for the lack of clarity in our article. We intended to use biotin-labeled downstream primers to verify the characteristics of the three differently labeled downstream primers. We have synthesized downstream primers labeled with Tamara and digoxin. We have revised the language expression, and the relevant content has been updated in the manuscript.

- Line 362- 366: should be removed

Response: Thank you for your suggestion. This section provides a necessary description of the sensitivity experiment, and we have moved it to the next paragraph to clarify *Broussonetia papyrifera*. We have revised the relevant content in the manuscript.

- Line 367: The title should be sensitivity only instead of specificity and sensitivity

Response: Thank you for your suggestion. We conducted specificity testing on both known positive and negative samples of BCoV, IBRV, and BVDV (lanes 1 and 2 in Figure 6 A), and the results revealed nactivity responses among BCoV, IBRV, and BVDV, with good specificity. We have revised the relevant descriptions in the manuscript.

- Line 370: The title should be changed; ..RRA using clinical samples

Response: Thank you for your suggestion. This section provides a necessary description of the sensitivity experiment, and we have moved it to the next paragraph to clarify the structure of *Broussonetia papyrifera*. We have completed the revisions to the relevant content in the manuscript.

- Line 378-380: should be removed

Response: Thank you for your suggestions. We have revised the language accordingly, and the relevant content has been updated in the manuscript.

Discussion

- Line 394- 410: Recast and send to introduction

Response: Thank you for your valuable suggestions. We have thoroughly revised the Discussion section of the manuscript, with a particular focus on in-depth analysis of the advantages, limitations, and challenges of our developed triplex

RAA-LFD technique. The corresponding modifications have been incorporated into the manuscript.

- Line 411-415: Introduce your overall result finding and discuss I therefore suggest you recast this whole paragraph.

Response: Thank you for your valuable suggestions. We have thoroughly revised the Discussion section of the manuscript, with a particular focus on in-depth analysis of the advantages, limitations, and challenges of our developed triplex RAA-LFD technique. The corresponding modifications have been incorporated into the manuscript.

- Lin3 415- 448 Discussion needs to be rewritten taking into account your results and findings

Response: Thank you for your suggestions. We have rewritten the content of this paragraph to highlight the overall research findings and conduct relevant discussions.

- Line 463: add 'the' after analysis

Response: Thank you for your suggestions. We have revised the language accordingly, and the relevant content has been updated in the manuscript.

Overall, we found that the reviewers' comments were quite helpful, and we revised our paper point by point. We earnestly appreciate the editors' and reviewers' work and hope that the corrections will be met with approval.

Once again, thank you very much for your comments and suggestions.

Re: Spectrum01628-25R1 (Developing a Recombinase-aided Amplification Method Combined with a Lateral Flow Dipstick Assay for Rapid Triplex Detection of Bovine Coronavirus, Infectious Bovine Rhinotracheitis Virus, and Bovine Viral Diarrhea Virus)

Dear Dr. Xuelian Ma:

Your manuscript has been accepted, and I am forwarding it to the ASM production staff for publication. Your paper will first be checked to make sure all elements meet the technical requirements. ASM staff will contact you if anything needs to be revised before copyediting and production can begin. Otherwise, you will be notified when your proofs are ready to be viewed.

Sincerely,
Alexander Bello
Editor
Microbiology Spectrum

Reviewer #3 (Comments for the Author):

The manuscript looks great, and the method developed will be handy in the field, saving costs.

I would like to suggest that the date brackets(months) of sampling within the two years be included.